# Recent Developments on Drivable Area Estimation: A Survey and a Functional Analysis

**DOI:** 10.3390/s23177633

**Published:** 2023-09-03

**Authors:** Juan Luis Hortelano, Jorge Villagrá, Jorge Godoy, Víctor Jiménez

**Affiliations:** Centro de Automática y Robótica, CSIC—Universidad Politécnica de Madrid, Ctra. Campo Real, Km 0.200, Arganda del Rey, 28500 Madrid, Spain; jorge.godoy@csic.es (J.G.); victor.jimenez@csic.es (V.J.)

**Keywords:** autonomous vehicles, drivable area estimation, road estimation, perception, computer vision

## Abstract

Most advanced autonomous driving systems (ADS) today rely on the prior creation of high-definition maps (HD maps). This process is expensive and needs to be performed frequently to keep up with the changing conditions of the road environment. Creating accurate navigation maps online is an alternative to reduce the cost and broaden the current operational design domains (ODD) of modern ADS. This paper offers a snapshot of the state of the art in drivable area estimation, which is an essential technology to deploy ADS in ODDs where HD maps are limited or unavailable. The proposed review introduces a novel architecture breakdown that fits learning-based and non-learning-based techniques and allows the analysis of a set of impactful and recent drivable area algorithms. In addition to that, complimentary information for practitioners is provided: (i) an assessment of the influence of modern sensing technologies on the task under study and (ii) a selection of relevant datasets for evaluation and benchmarking purposes.

## 1. Introduction

The last decade has seen autonomous vehicles take their first steps out of academia into the commercial world with some companies starting to offer completely unmanned taxi services in selected American cities. These services are usually bound to the prior generation of HD maps. These digital maps are highly precise representations of the driving scene containing centimeter accurate descriptions of the road shape, driving lanes, traffic markings and traffic signs. The creation and maintenance of HD maps stands as a barrier for the commercial viability and general deployment of autonomous vehicles because it is a time-consuming and expensive operation [1]. HD maps are usually captured with an extensive array of sensors, such as GPS, 3D LiDAR, HD RGB cameras, radar and aerial imaging, that then need to be processed offline and stitched to existing sections of the map. The output of this process is a static map that is sensitive to the everyday variability of the drivable area [2]. An alternative to this process is to build accurate online maps in the autonomous vehicles themselves while they are out operating.

Building navigation maps online requires several highly complex tasks performed at once: high precision localization, online identification of traffic markings and signs and an accurate and fast drivable area estimation [3]. This publication focuses on the latter by drawing a picture of the state of the art in drivable area estimation in regards to algorithms, sensors and datasets. The main contributions of this survey are:An analysis of the state of the art of the last eight years in regards to drivable area estimationAn architecture breakdown and a taxonomy that fit learning-based and non-learning-based algorithmsA study of the existing relevant datasets to assess the performance of these algorithms and the influence of modern sensing technologies in their performance.A proposal for future research directions for the field

The rest of the paper is structured as follows: Section 1.1 focuses the scope of the paper in relation to its adjacent research fields and Section 1.2 presents other relevant surveys. Section 2 presents the challenges of drivable area perception with a brief description on especially adverse situations for sensors. Section 3 describes the datasets that are useful for the research and development of algorithms in the field. Section 4 presents the description and analysis of the chosen algorithms plus the proposed architecture breakdown and taxonomy. Section 5 tries to synthesize the findings into a proposal for future research directions, and finally, Section 6 draws some concluding remarks.

### 1.1. Scope

In this work, drivable area is defined as the area that is physically accessible and designed to be traversed by vehicles. The publication main objective is to study the techniques and the literature works that allow an autonomous vehicle to differentiate road from sidewalks, curbs and other non-navigable ground areas. Publications that, on top of estimating the drivable area, offer additional information such as lane-lines are also considered. The estimation of traffic rules from traffic signs and markings is considered out of scope for this publication.

Drivable area estimation can also be considered a subset of larger segmentation tasks such as semantic, instance or panoptic segmentation. Semantic segmentation tries to label every pixel of an image or point in a pointcloud into the clusters that belong to the same class. Instance segmentation produces a specific label for each occurrence of the same object in the scene but leaves out the background if it cannot be assigned to an object. Panoptic segmentation combines semantic and instance segmentation in order to produce a differentiated label for everything that can be seen in one view of the scene.

Detecting the drivable area has specific problems and characteristics that makes it different from segmenting other classical segmentation labels such as cars, pedestrians or traffic signs. The drivable area is usually locally static, and it is located underneath the vehicle and surrounding it. It needs to be detected constantly since the absence of it should cause an immediate emergency stop. Its appearance varies in color, texture and shape with no standard form. Furthermore, opposed to some other labels, the drivable area can be merged with offline registries such as in low fidelity maps.

Studying this problem in isolation allows for the creation of an architecture breakdown of the problem that sheds light on how each stage of the strategies affects the detection of the drivable area. This analysis is significantly different than using holistic segmentation techniques, which may hinder the understanding of how each stage affects each label. Even though some of the datasets will be shared by the study of the different disciplines, the proposed algorithms and functional analysis are distinct enough to warrant a stand-alone survey.

### 1.2. Related Surveys

Drivable area estimation has been studied in other high impact surveys that treat the problem with varying degrees of specificity. In [4], the authors offer a thorough review of the state of the art of the 2000s, an exhaustive and fitting architecture breakdown for non-learning-based techniques and the identification of gaps in the research. The future research directions they suggest have proven very inspired, as all of them were explored in the following decade: the creation of public benchmarks, the application of machine learning techniques and the implementation of sensor fusion. In [5], the author presents a survey of terrain traversability for unmanned ground vehicles that offers an interesting perspective on the ground variability that a vehicle could encounter, which helps broaden the focus from standard roads. The authors in [6] published a modern survey that presents learning-based lane estimation methods with a focus on ego-lane detection. Finally, the authors in [7] present a very comprehensive survey of learning-based 3D LiDAR semantic segmentation that is useful for the subtask of drivable area estimation because it specifically provides a thorough review of the current datasets and discusses how those data are currently generated and consumed.

The present survey collects publications without discriminating between learning-based and non-learning-based methods with a focus on drivable area estimation plus an analysis on the sensors and datasets needed to develop the state-of-the-art research in the field.

## 2. Drivable Area Perception

Drivable area estimation demands a specific performance that the sensors achieve or miss depending on the situation. It is the norm to mount an arrangement of different sensors in the car since it is understood that one single sensor type would not adequately adapt to the myriad different scenarios in real-world driving. A brief performance analysis of each sensor relative to drivable area estimation is presented below.

### 2.1. Rgb

RGB cameras are a constant bet by autonomous driving researchers and industrial developers due to its low cost, size and technology maturity. They are also the closest sensing technology to human sight, which explains why most of the traffic language is visual and, therefore, adapted to cameras. Figure 1 shows an example RGB view of a roundabout entrance using the camera of a Google Pixel 4a with a resolution of 4032 × 3024.

The challenge of using RGB cameras is that the exact same visual traffic information can have very different sensor values depending on external factors. Figure 1a,b show how going from day to night and from dry to wet conditions changes the values for color, intensity and reflectance of the image. These differences can even appear on the same image due to shadow and illumination changes.

### 2.2. 3D LiDAR

3D LiDAR is the technology that has seen the biggest improvement in the past decade. The progress has come as an increase in the number of layers, point density, reflective sensitivity and output frequency. Figure 2 shows an example road view of three state of the art 128-Layer 3D LiDARs. These sensors display a dense representation of the surroundings that allows for a clear road perception. It is interesting to see how those three top-of-their-line sensors differ in their reflective sensitivity; in Figure 2b, the sensor is able to give very high reflectivity values for the lane lines and the vegetation, making the road more apparent. The sensor in Figure 2c is able to pick up on the lane markings but struggles with vegetation. In Figure 2d, the sensor is able to discern between road, vegetation and lane markings in reflectivity alone but misses less reflective road points that are further away from it.

Even though the technology has taken a considerable step forward, it still has its drawbacks when tackling the drivable area estimation task. Some of these shortcomings can be seen in Figure 3. Figure 3a,b show how a high-end 3D LiDAR can suffer data loss from poorly reflective surfaces such as black vehicles. Figure 3c,d illustrate how it can be hard to discern between pathways and drivable area in 3D LiDAR pointclouds.

### 2.3. Open Source Maps

Another common way of acquiring information about autonomous vehicle surroundings is through the use of open source maps. The most widely adopted provider is Open Street Maps (OSM) [8], which is a crowd-sourced repository of map features that contains geolocalized road topology, signals, traffic rules and buildings. This source of information can be useful to tackle the drivable area estimation task since it contains geospacial limits to the road, several works already make use of it [9,10]. However it cannot be a complete and standalone solution due to the potential errors introduced in the map caused by manual data acquisition and entry and due to the fact that the rate at which the environment changes is higher than the rate at which users update the open database. Figure 4a shows a section of a roundabout stored in the OSM database. Figure 4b shows the OSM data from Figure 4a transformed into the Lanelet2 [11] format in order to process each individual lane by using the Commonroads open-source OSM-Lanelet2 converter [12].

Figure 5 shows some of the errors and drawbacks of open-sourced maps. In Figure 5a the difficulty in aligning the lanes derived from OSM with satellite imagery in some specific scenarios can be seen. These issues are similar to the ones that may arise when aligning the images with the GPS position from an autonomous vehicle. Figure 5b shows that the number of lanes stored in the OSM database does not always match the current state of the road.

## 3. Datasets

As can be seen in previous surveys [4], one of the biggest gaps in the state of the art was the lack of dedicated datasets that would help to initiate, validate and compare research in the drivable area estimation task. This section presents a set of datasets that have emerged in the last ten years and have proven to be especially useful to tackle the problem at hand. Table 1 presents a summarized view of those datasets. The exclusive criteria used to choose them was modernity, public availability and applicability to the problem, which is characterized by presenting urban scenes with a wide choice of sensor configurations. The non-exclusive criteria was the presence of benchmark mechanisms (BM), which can be leaderboards or competitions that allow for public comparison, the inclusion of geolocalization to potentially merge them with other sources of information, sequentiality, separate testing and training datasets and the relevance of the publications that make use of them. Below, there is a description of the selected datasets based on their applicability to the problem.

**Waymo Perception** as a part of the Waymo Open Dataset is one of the most significant sensor datasets in terms of the sheer quantity and variety of annotation types. On top of its sensor suite, it offers labels for bounding box, key point, 2D panoptic segmentation and 3D semantic segmentation. It shows urban and residential scenes with diverse weather in both daytime and nighttime environments. It was updated in 2023 with an HD Map, and it also includes a python devkit to streamline development. **Argoverse 2** builds its value proposal on top of the long range (200 m) of its LiDAR and annotations, the six different US cities it collected its data in and the amount of different labels it offers (30). It also offers a vector map with lane-level geometry and a ground height raster map to ease the filtering of ground LiDAR returns. **nuScenes** is a dataset that implements a full sensor array with RGB cameras spanning a 360° view, a 32-layer 3D LiDAR and a RADAR array. It includes a high number of annotations in inner-city traffic scenes with changing weather and heavy traffic. It has a leaderboard that differentiates between LiDAR only and any sensor modality for semantic segmentation and panoptic segmentation. **CityScapes** is a Stereo RGB dataset of complex urban scenes that offers fine and coarse semantic annotations. It depicts busy inner-city driving with high levels of traffic and pedestrian interaction and has separate annotations for road and sidewalk. **KITTI** is one of the oldest and most popular autonomous driving perception datasets available. The specific subset for drivable area estimation is named KITTI ROAD, and it consists of a non-sequential set of images and pointclouds taken in non-busy suburban environments with and without lane markings in single- and multi-lane roads. It has become the standard benchmark for the task. and one of its main strengths is the dedicated leaderboard, which acts as a powerful surveying tool of the current state of the art. Its main drawbacks are that it has no weather or illumination diversity and the relative sparsity of its traffic scenes. **Semantic KITTI** spawned as a complementary dataset from KITTI’s popularity and it expands on its features by adding semantic annotations for all sequences of the odometry subset. **BDD100K** is an extensive RGB-only dataset that offers 100.000 sequential images of driving in four different American cities. This dataset offers lane marking and drivable area annotations, making a distinction between directly drivable and alternatively drivable areas. They use (i) the directly drivable tag for road areas in which the ego-vehicle is currently driving on and has the priority, and (ii) the alternative drivable tag for road areas that the ego-vehicle is not currently driving on but could through a lane change. This dataset also offers a dedicated leaderboard based on yearly competitions. **KITTI-360** calls itself the successor of KITTI and expands on the latter by offering richer sensor modalities, semantic instance annotations and a more accurate localization in suburban scenes with moderate traffic. It expands on the sensor suite of the original KITTI by adding an additional LiDAR, a pair of front-facing RGB cameras to produce disparity maps and two lateral fisheye cameras to complete their 360º scene perception. The main value proposition of this dataset is that it is the only segmentation-oriented dataset with all sensor modalities simultaneously available. **DIODE** is a combined indoor/outdoor dataset that offers RGB, depth and normal information using the same sensing and imaging setup. It achieves this by using the FARO Focus S350 sensor, which is an actuated phase-shift laser scanner that creates RGB and depth scenes with very high accuracy, resolution and FOV. The downside of using this dataset for drivable area estimation is that it has no sequential scenes as all of them are static due to the nature of the sensor. **3DHD CityScenes** is a dataset that combines high-definition maps with high-density synchronized and georeferenced pointclouds taken by a high-end spatial imaging sensor. **OpenLane V2** is a dataset built on top of nuScenes and Argoverse 2 that focuses on scene structure perception and reasoning by offering a dynamic map that takes into account traffic elements such as ground markings and traffic lights and signals. One of its highlights is the fact that they offer 3D-annotated lanes in their map, as opposed to the 2D lanes present in most datasets. **Online HD Map Construction Benchmark** offers a set of vectorized and rasterized maps from camera images that is built on top of the nuScenes dataset.

## 4. Drivable Area Estimation

This section presents a review of the state of the art in drivable area estimation methods and the architecture breakdown, plus the taxonomy that can be inferred from them. The architecture breakdown and taxonomy are presented first in an attempt to provide a framework to analyze the algorithms that can be seen in the latter part of the section.

### 4.1. Architecture

A thorough study of the literature unveils a set of common stages for most of the drivable area estimation algorithms. Figure 6 presents an architecture breakdown based on the similarities found in the shared modules of the algorithms. It is important to note that this diagram depicts a generic architecture and not all works employ every stage or necessarily follow the same order. Nonetheless, it is possible to map most of the research to a subset of this architecture breakdown. In the following, there is a description of each stage.

**Noise Removal** is the process of identifying and correcting sensor data with the purpose of improving algorithmic or computational performance. Two types of noise are identified: static and motion noise. Static noise encompasses data that are irrelevant or harmful to the algorithm, such as outliers or out-of-range information, and are inherent to the sensor. The removal process of this type of noise can range from simple and fast through thresholding processes, such as min–maxing points that exceed a set height value, to more complex and costly, such as plane fitting through least-squares or RANSAC. Motion noise causes deformations in relevant information to the algorithm and is created by the vehicle’s movement. Example correction processes for this issue are pointcloud deskewing or attitude alignment. **Modal Transformation:** As a previous step to fusion or in order to apply cross-field techniques, a modal transformation to sensor data can be performed. Modal transformations usually affect data dimensionality by means of projecting (e.g., 3D LiDAR pointclouds onto image coordinates) or deriving (e.g., creating a map of normals from an image or a pointcloud). **Fusion** is the process of combining data coming from different sensors or stages to increase the accuracy that would be achieved in isolation. It is a stage that can be performed at different times during an algorithm execution as it can be applied to sensor information or to already generated features. Common fusion techniques are grid maps and Kalman filtering. Fusion inputs can be diverse: raw sensor data, processed sensor data, low-level features and high-level features. **Feature Extraction:** The translation from high-dimensional rich sensor data into basic features takes place in feature extraction. Features useful for drivable area estimation can be low level, such as image color or texture, or high level, such as detected lane lines in RGB images or height differentials extracted from pointclouds. **Feature Expansion** formulates hypotheses of the drivable area and fits them to the available extracted features. It generates drivable area proposals and uncertainty estimations. Applicable techniques in this stage are image upscaling, graph search or model fitting. **Tracking:** Using an estimation of vehicle displacement and geospacial information, this stage matches the produced estimations through time in order to improve the output. Tracking can be achieved by applying Bayesian fusion or Kalman filtering. **Neural Network** deep learning technologies have proven useful to act as a backend in any of the drivable area estimation stages. Neural networks can be trained to perform any stage in isolation and also to perform several tasks at once or even the complete algorithm from start to finish. Network training can be performed at different points in the algorithm’s timeline while taking a variety of different inputs (e.g., raw sensor data, data after modal transformation, data after fusion, features, etc).

### 4.2. Taxonomy

Drivable area estimation is now a mature field that has spawned a varied set of techniques that can be applied to tackle the problem. The analysis of this field enabled the synthesis of a taxonomy that can be seen in Figure 7. It is important to note that a complex problem such as drivable area estimation usually requires the application of several techniques at once, and therefore, those sets are not exclusive, and it is common that one set of techniques feeds off another. A description of each family of techniques is provided below.

**Constraints.** Drivable area estimation deals with the detection of man-made, industrially designed artifacts such as roads, curbs, sidewalks, etc. As such, it is useful to limit the search space using carefully chosen constraints. Symmetry constraints are commonly both finding features and fitting models, as most roads are defined by left and right boundaries that are also parallel. Finding the twin equivalent of a feature or constraining a model to remove outliers is a sound geometric foundation for road detection. Smoothness constraints. Roads are made for non-flexible, bound-to-the-ground and several hundred kilos heavy vehicles that cannot make abrupt direction changes and, therefore, require gradual evolution to be traversable. Smoothness constraints help find drivable area estimation candidates and predict its evolution. Continuity constraints are useful to find matches between consecutive sensor reads since drivable areas do not usually abruptly end or change. Fixed size constraints. Most roads maintain a fixed width that can be bound between a maximum and a minimum value that is useful to weed out outliers and find equivalents when also applying the symmetry constraint. Flatness constraints are useful because surfaces need to be locally flat when underneath a vehicle, or otherwise, at least one of the wheels would not be in contact with the ground. **Features** are acquired by processing raw data coming from the sensors and creating data points that host new information. Color features use pixel properties in RGB images to differentiate the drivable area from other vehicles, pedestrians or non-drivable surfaces. They are sensible to lighting changes, adverse weather or similarly colored bodies that do not belong to the same category. Edge features try to identify the drivable area by detecting hard gradients in color, intensity or geometry, as those are commonly found in the road limits with sidewalks or ditches. Texture features extract information from how color is arranged spatially in an RGB image and help produce drivable area candidates on the assumption that texture patterns remain consistent within the same road objects. Normal features are produced by calculating the normal vector of a group of geometric points and clustering them based on their angle. Usually, road normals point in the same direction or change gradually. Reflectivity features can be acquired using modern LiDAR sensors and give information about how any surface is able to reflect or absorb light. It is useful to detect road markings and signs as they are usually highly reflective. It could even be used to detect the road itself as a poorly reflective area if the sensor is sensitive enough. **Modeling** fits the sensor data to mathematical models that define lines or planes. Straight lines are a common model that is cheap computationally and can be directly applied to simple roads or as a collection of segments to model complex roads. Splines are smooth piecewise functions defined by polynomials. They have variable complexity depending on their order and, therefore, are useful to model curves. Bezier curves are smooth global functions defined by polynomials. They are attractive as a model because they offer curvature continuity at every point and are not computationally expensive to compute in their closed-form expressions. Polynomials of the second or third degree are useful to model curves under some constraints (p.e flatness) as they are simple to understand and compute. Their drawbacks are that modifying any points affects the complete curve or undesirable effects at the boundaries. Planes are especially useful in LiDAR data processing as planes are a very common feature in LiDAR pointclouds that appear in roads, sidewalks or buildings.All of these model proposals need to be fitted to the data using a mathematical approach. A common method to estimate the parameters of a model is random sample consensus (RANSAC), which consists of iteratively checking the fitness of a random sample of the data against a previously set model. It is useful to separate inliers from outliers but is a non-deterministic model, which means that its accuracy depends on the amount of iterations it has run. Another technique is principal components analysis (PCA), which is a method for dimensionality reduction that tries to keep most information in the dataset. It linearly transforms the data into a new coordinate system where the majority of their variance can be described using less dimensions than the original data. The challenge in applying PCA in this field is usually selecting the data subset in which the model needs to be fitted. Finally, least median squares (LMS) tries to minimize the sum of the squared difference between an observed data point and the fitted value provided by a model. It is a simple method to apply and understand but is sensitive to outliers and can overfit the model. **Representation** methods offer alternative representations for sensor and feature information that reduce dimensionality and ease the computational load. They are also widely used for data and feature fusion. Occupancy Grids represent the world with evenly spaced cells that host a binary variable (occupied or free) that is estimated and assigned a probability. These grids are used to make drivable area assumptions over the free area. Elevation Map models the world by keeping height information that can be used to derive the drivable area by detecting abrupt height gradients. Polar Grids host the information in polar coordinates instead of in Cartesian coordinates. They usually result in smaller and, therefore, faster to compute grids, have higher resolution near the vehicle and are adequate representations of roads because they both grow radially from the vehicle. Triangle Grids are common world representations in fields such as videogames because they can accurately model height and complex shapes. Their main characteristics are planarity, which means that every point of a triangle can be at different heights and still belong to the same plane, and simplicity, since they have the lowest amount of vertices of any polygon. **Propagation** techniques create relational systems between data points to mitigate the issue of data gaps caused by occlusions or at great distances from the sensor. Markov random field is an undirected and cyclic graph technique that sets each data point as a node and tries to assign labels and their probability to them by performing inference techniques on the graph, such as belief propagation. Conditional random field is a special case of Markov random field in which the graph takes into account the influence between neighboring nodes by modeling the dependencies between nodes. Bayesian Generalized Kernel propagates information by assuming continuity between adjacent data points and inferring the missing information by applying a kernel to the neighboring observed points. Dempster–Shafer theory is a general mathematical framework to reason with uncertainty and deal with information gaps. It is designed to work with sets of different labels combining evidence from different sources and producing a degree of belief. **Learning-Based** methods are ubiquitous due to their flexibility that allows them to morph into several different techniques. They can have sensor data, features or grid representations as inputs and can be used to perform model fitting, feature extraction or generate drivable area candidates directly. Their main drawback is the need for large annotated datasets for training. Convolutional Neural Networks are very popular learning-based approaches in the field. They are usually applied directly to RGB images from the road or to images derived from LiDAR pointclouds. They work by training image kernels that learn to identify specific features in an image and then passing those features to a fully-connected layer that recognizes larger elements in the scene. Residual Neural Networks are a special case of convolutional neural networks that, at the same time, try to reduce the number of layers of the network and tackle the vanishing gradient problem that occurs in deep networks. They work by providing residual connections between skipping layers that avoid the activation functions of previous layers that would reduce their derivatives. This results in shallower networks that are more efficient to train.

### 4.3. Algorithms

This section presents the set of algorithms that describe the current state of the art, their functional analyses and their contributions and limitations. The algorithms and their analyses are collected in Table 2. The criteria for choosing this specific set of algorithms was: (i) *scientific relevance*, the impact that this set of algorithms has in the state of the art and the novelty of their contributions; (ii) *ensuring a broad representation of techniques*, making sure that this survey presented a set of algorithms that covered every type of sensor combination and showed a plural application of the available theory; and (iii) *benchmark appearance*, including the highest performing algorithms today in terms of drivable area estimation quality measures and computation time.

The diagrams appearing in the architecture column depict the functional analysis of each algorithm in a simplified schematic manner by fitting them into the stages appearing in Figure 6. For reference, 
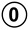
 is neural network, 
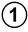
 is noise removal, 
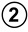
 is modal transformation, 
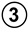
 is fusion, 
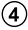
 is feature extraction, 
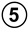
 is feature expansion and 
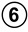
 is tracking. Algorithms that show several branches describe works that perform stages in parallel.

The neural network stage depicts the moment of training the network in the algorithm and the inputs it receives to do so. Once this stage appears in an algorithm, all subsequent stages are then performed by neural network inference. For example 
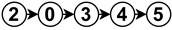
 implements modal transformation and then uses this information to train a neural network that performs fusion with raw data, feature extraction and feature expansion.

Following, there is a description of each selected publication’s stages and methodology:**BGK [25]** first performs coarse ground segmentation through a min–max height difference approach and projects it into a grid. Then, they estimate the missing height information in the grid through Bayesian generalized kernel inference. From the generated dense height grid, they estimate the normal vectors and compute their angle difference to produce navigable candidates through angle thresholding. Afterwards they apply bilateral filtering to preserve edges, and finally, they apply multi-frame tracking on consecutive grids. **SNE-RoadSeg [26]** estimates surface normals from the depth information coming from a RGB-D sensor using a surface normal estimation (SNE) module. Then, it feeds the estimated surface normal data and the raw RGB to a convolutional neural network (CNN) with a parallel encoder–decoder architecture with densely-connected skip connections. 

**PLARD [27]** performs an altitude difference-based transformation (ADT) to produce an altitude difference image from the LiDAR data. Then it inputs the RGB data plus the produced altitude difference image into a CNN that fuses the generated features through feature space adaption (FSA) in its intermediate stages to finally produce a road estimation candidate. **USNet [28]** performs surface normal estimation from the depth information coming from the RGB-D sensor. It then feeds the estimated data and the raw RGB data into two separate CNNs that generate feature maps. Finally, those feature maps are fed to a module called multi-scale evidence collection (MEC) that generates separate prediction and uncertainty maps. **Unsupervised RD [29]** fuses RGB and LiDAR data to create the superpixel data structure. Then, it performs Delaunay triangulation to assign spatial surface to the superpixels. From the estimated surfaces, it computes its normals and uses them to estimate obstacle points through flatness edge thresholding. Then, it creates a ray map by casting rays from the origin of the sensor to the detected obstacles, which produces a rough road estimation. Finally it uses Markov random fields(MRF) and belief propagation to perform feature fusion with the objective of increased robustness. **Map-Supervised RD [9]** generates training annotations using OSM and geo-referenced images from the KITTI dataset. It then refines the generated labels by clustering pixels with similar color. Finally, it then uses the automatically generated labeled data to train a CNN. **RBANet [30]** uses SegNet [50] to create residual feature maps and then apply reversed attention (RA) and boundary attention (BA) units to them to generate road estimations. **HID-LS [31]** fuses RGB and LiDAR data to create superpixels. Then it performs Delaunay triangulation to interpolate the missing gaps in the 3D information. In the generated data, it computes inverse depth and height maps. It then uses the inverse depth map to compute horizontal and vertical histograms and derive first road region estimation. Using row and column scanning in the height map, it derives the second road region estimation. Finally, it fuses both estimations to produce a final road candidate. **Curb Detection [32]** first applies RANSAC to extract the ground plane with sidewalk, curb and road points. Then, it performs sliding-beam segmentation to divide the extracted ground areas in regions of interest. Finally, it extracts features through angle thresholding to detect curbs. **LiDAR-Histogram [33]** creates LiDAR imagery by displaying the points in a 2D plane organized by pitch and rotation angle. Then, it computes histograms from the LiDAR imagery. Afterwards, it performs RANSAC line fitting on the histograms to estimate road lines. Finally, it segments the pointcloud based on the estimated road lines. **CyberMELD [10]** computes vertical and horizontal slope maps and sums them to, afterwards, grow and detect a general region of interest (ROI) road area. It then performs Delaunay triangulation to map ROI 3D Points onto the RGB image. In the generated data, it applies inverse perspective mapping (IPM) and finds lane lines on it using gradients. Finally, it fuses the lane lines with OSM to generate an ego-lane estimation. **RoadNet3 [34]** uses cascaded CNNs to reduce the resolution of the feature map followed by a long–short-term memory (LSTM) network to find the contour of the drivable area. It applies the same architecture in parallel in the full image and in a smaller region in the center of the image and fuses them to produce the road estimation. **Double Projection [35]** creates a 2.5D elevation map with sets of points falling under the same cell through a min–maxing process. It also creates a 2.5D range map by projecting LiDAR points to a virtual cylinder plane. Then, the elevation map is used to detect obstacles through thresholding, and a reachable area is estimated through forward flood fill. The range map produces a road area using a smoothness constraint. Finally, the reachable area and the road area are fused through Bayesian decision theory to extract the drivable area. **Pseudo-LiDAR [36]** uses a big to small [51] neural network to estimate depth from RGB. It then transforms depth into attitude space to generate a Pseudo-LiDAR [52] pointcloud. It then applies a residual neural network (RNN) to process features in parallel for RGB and Pseudo-LiDAR and fuse them at different points of the pipeline to obtain the final result. **CLCFNet [37]** transforms a LiDAR pointcloud into LiDAR imagery. Then, it performs a perspective transformation to put the LIDAR imagery in the camera view. Then, it inputs the LiDAR imagery, LiDAR pointcloud and RGB into three cascaded CNNs to extract and fuse the road features. It works with LiDAR only or with both LiDAR–RGB data depending on light conditions. **Multi-Cue [38]** computes normal vectors on the disparity maps from stereo images. It then finds interest boundary areas from highly diverging normals. Curve fitting on the boundary pixels by using support vector regression(SVR) [53]. **TRAVEL [39]** first corrects pointcloud skew and attitude caused by ego-motion. Then, it models the terrain by grouping subsets of pointcloud points into tri-grid field nodes. Afterwards, it uses breadth-first traversable graph search to classify traversable nodes by measuring acceptable concavity and convexity. Finally, it applies model fitting to match the traversable nodes to pointcloud points, assigning them a label. **Road Markings [40]** implement coarse ground segmentation through RANSAC plane fitting, along with regional grow-clustering to weed out points belonging to the curb. Then, they apply adaptive thresholding based on Otsu’s method [54] on the reflectivity information coming from the sensor. Finally, they produce lane boundary proposals by line model fitting. **Line Fitting [41]** first implements coarse ground segmentation through the channel-based clustering of points in a 2.5D polar grid map, along with height thresholding. Then, it identifies boundary points by checking the angle, distance and height difference between adjacent points. Finally, it performs B-spline curve fitting to produce the road boundary candidate. **YOLOP [42]** uses CSPDarknet [55] as a backbone to extract feature maps. Then, it implements spatial pyramid pooling (SPP) [56] to fuse features of different scales and a feature pyramid network (FPN) [57] to fuse features of different semantic levels. Then, it uses an upsampling process to restore the original size of the image from the feature map and generate two separate segmented images of the drivable area and lane lines. **HybridNets [43]** is a neural network that uses EfficientNet-B3 [58] as a backbone to extract feature maps and an FPN as neck to fuse features across feature maps with different resolutions. Then, it upsamples feature maps up to half the original resolution and feeds it to its proprietary segmentation head to to produce the final multi-label segmented image. **Rangenet++ [44]** performs a spherical projection of the LiDAR pointcloud into a range image and then feeds it into a fully convolutional neural network with an encoder–decoder hour-glass-shaped architecture. Finally it performs a k-nearest neighbor (kNN) search to reduce noise and shadow-like artifacts in the produced multi-labeled output. **Urban Road Filter [46]** tries to detect the road by applying three different techniques to LiDAR pointclouds. The three different techniques differ in the way they divide the pointcloud, by channels, by beams and by sliding windows. It then applies different heuristics to the clusters to detect anomalies in height or angle differences to produce curb candidates. Finally, those curb candidates are used to produce a drivable area polygon. **Evidential Grids [47]** use the Dempster–Shafer theory to efficiently fuse information sources that provide partial information of the environment. It offers a framework to fuse occupancy grids created from LiDAR information with meta-knowledge obtained from a high-fidelity map. It outputs drivable and non-drivable space and can differentiate from stationary and moving objects through temporal fusion. **RoadSLAM [48]** separates the pointcloud into ground and non-ground through coarse segmentation and clusters it in sets of free areas. Then, a robust weighted least-squares curve fit is applied to each side of the selected free area in order to find the instance that maximizes the likelihood given the current route of the vehicle. An unscented Kalman filter (UKF) is used to produce a prediction of the control points of the B-Splines representing the road boundaries with the free area boundaries as input. Finally, all information is accumulated and fused using GraphSLAM [59] with OSM as prior information. **SemanticDepth [49]** creates two information sources from a single RGB input. First, through monocular-depth estimation, they generate a disparity map from a single RGB image and then they transform it to a full pointcloud. Then, they perform semantic segmentation on the original RGB image using a CNN. Then, they overlay the segmented image as masks on the generated pointcloud to calculate the road width ahead of the ego-vehicle. 

Table 3 presents the taxonomical classification of the algorithms. The categories in the column are based on the study presented in Section 4.2.

The acronyms used in the table are the following: For the constraints group: symmetry (Sym), smoothness (Smo), continuity (Con), fixed size (FS) and flatness (Flat). For the Features group: color (C), edge (E), texture (T), normal (N) and reflectivity (R). For the model group: straight line (SL), splines (Sp), polynomial (P) and planes (Pl). For the representation group: occupancy grid (OG), elevation map (EM), polar grid (PG) and triangle grid (TG). For the propagation group: Markov random fields (MRF), conditional random fields (CRF), Bayesian generalized kernel (BGK) and Dempster—Shafer Theory(DST). Finally, for the learning group: convolutional neural network (CNN) and residual neural network (RNN). The Bezier category present in Section 4.2 was left out due to no selected algorithm making use of it.

Table 4 shows the algorithms performance as is reported in their publications or in the leaderboard where they appear.

The computational performance in the time column is shown in milliseconds per frame, representing the time it takes to produce one instance of drivable area estimation. The algorithmic performance is represented by the reported measurements of precision, recall, F-score, mean intersection over union (mIoU) and mean absolute error (MAE), with the exception of the three algorithms that did only include qualitative experiments in the publication. The GPU column denotes that the algorithm requires a dedicated parallel processor to reach the reported performance. The dataset column indicates which of their reported performance numbers were chosen to be included on the table. The code column indicates whether those publications offer access to their code. The criteria to choose between datasets whenever more than one set of performance metrics was available was the frequency of appearance in the other publications in order to facilitate fair comparison. Most of the publications included performance measured at least on the KITTI dataset, which explains its prevalence on the table.

Finally, some interesting remarks that can be inferred from studying Table 2, Table 3 and Table 4 are:There is, overall, high reported performance in all the publications. Every algorithm reports at least an F1 score of 83, with one of the methods even reaching 97.Execution time and performance are not clearly correlated as the highest performer has a runtime of 460 ms but a very close second only needs 22 ms.Learning-based methods usually offer the higher performance [26,27,28,30,36]. However, there are non-learning-based methods that reach similar performance levels [35,40,41].In terms of sensing capabilities, it seems that fusing semantic information plus depth (p.e. LiDAR or RGB-D + RGB) pays off in terms of performance. Nevertheless it is not a must as publications with a single sensor input also perform reasonably well [30,36,40].Neural networks offer consistently high performance at a relatively low computational cost but they all require a GPU to perform. The best tradeoff between computation time and algorithm performance is offered by USNet.With roughly 65% of the works under 100 ms in execution time and 47% under 50 ms, it is possible to see that a majority of the works today can handle sensor inputs at 10 Hz in real time and almost half can handle 20 Hz, which are two standard 3D LiDAR frequencies.

## 5. Discussion—Future Research

Drivable area estimation has taken a leap forward in the last decade with the advancements in sensor technology, computational power and algorithmic innovation. The maturity of learning-based methods has broken into this field and raised the bar in terms of computational time and performance measures. Three-dimensional LiDAR technology has seen an increase in data quality, resolution and frequency while lowering its cost, finally making it a viable option for researchers and industrial players. Dedicated computational platforms with integrated GPUs have appeared in the market and are used as the processing core of autonomous vehicles. All of this paints a thriving image of the field today, which, nonetheless, still has some threads to pull until it becomes a solved problem:**Fusion variability:** Sensor technology advances have brought the importance of sensor fusion to the foreground. Different sensors fill gaps in the drawbacks of others, and having a diverse array has become vital to reach a robust solution. Most of the methods currently tackle this by optimizing an algorithm for a sensor type and then modal transforming the sensor data from a different type into the optimized one. A potential research direction that would increase performance is delaying modal transformation and fusion to later stages of the algorithm and to optimize separate parallel algorithms for each sensor type. The authors in [37] propose a study with results from different fusion architectures that follows this direction.**Three-dimensional LiDAR reflectivity:** Modern developments in sensor technology have brought to the market 3D LiDARs with more sensitive receptors to light reflection intensity. The technology has come to a point where road shape can be understood from reflectivity alone. The drivable area estimation algorithms reliant on reflectivity features could pose a new breakthrough in the state of the art. The authors in [40] introduce a technique for lane marking estimation based on lidar reflectivity that could be adapted for drivable area estimation.**Low-fidelity map fusion:** Open-source low-fidelity maps offer valuable context information for drivable area estimation algorithms that is currently underrepresented in the state of the art. Some of the map providers see their contributor base increasing each year [60], while new players backed by big tech companies are appearing now [61]. A research direction for the field could be algorithms that take advantage of those databases as a core part of their proposal. A publication that is already working in that direction is [10].**Streamlined data processing:** Technological advancements in the sensor industry have brought sensors that produce high-density and high-quality outputs. Learning-based methods have their performance tied to having access to a significant amount of data. Therefore, the field would greatly benefit from new data storage and processing systems that could streamline the flow of data.**RGB strongholds:** RGB-based drivable area estimation is being spearheaded by learning-based methods while LiDAR does not have a clear technique leader yet. A potential gap in the current research is novel learning-based methods that only use 3D LiDAR.**Algorithm output:** The algorithms express their road estimation differently in terms of data type and quantity. Depending on the desired application, some outputs could be more interesting than others. Some relevant criteria to study the algorithms could be: (i) if the algorithm outputs is single or multi-label, (ii) if the algorithm outputs an expression of probability or uncertainty together with the estimation, (iii) if the output type is an image, a map, a pointcloud or a road model. All of these can significantly influence the choice in research direction. One example could be the need for an uncertainty measure helping to deal with potential sensor errors. Another could be the fact that a segmented image is likely to need an extra processing step to obtain a general road representation while a road model can be directly used by the next module.**Multi-frame estimation:** Algorithms that fuse estimations from different timesteps are not very common. The few ones that combine estimations carry them out in tracking or recursive neural networks. A potential research direction is to apply multi-frame estimation to produce reliable and robust intermittent-noise drivable area candidates.

## 6. Conclusions

In this publication, a thorough study of the drivable area estimation literature is conducted in order to take a snapshot of the state of the art as it stands today. This investigation has produced an analysis of the modern sensors used to address the problem and their drawbacks, a summary of the most adequate datasets for research validation, a novel architecture breakdown that helps understand and analyze drivable area estimation algorithms, an outline of the most relevant algorithms in the field with a performance comparison and a suggestion for future research directions. Altogether, this publication aims to offer a snapshot of the state of the field at this point in time and an aid to find a path forward.

## Figures and Tables

**Figure 1 sensors-23-07633-f001:**
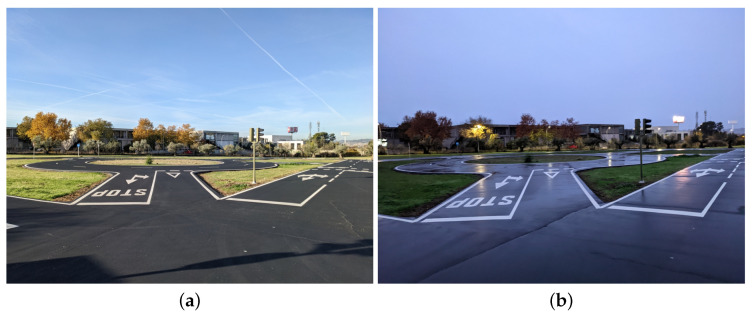
RGB view of a roundabout entrance. (**a**) Day view in dry conditions. (**b**) Night view in wet conditions.

**Figure 2 sensors-23-07633-f002:**
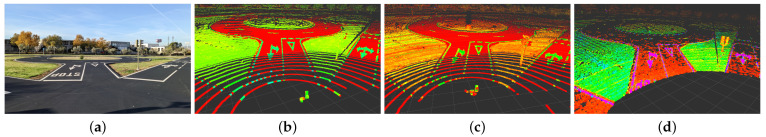
3D LiDAR view of a roundabout entrance color-coded by reflectivity (red is lower, green is higher). (**a**) Reference picture. (**b**) Robosense Ruby Plus. (**c**) Velodyne Alpha Prime. (**d**) Ouster 0S2.

**Figure 3 sensors-23-07633-f003:**
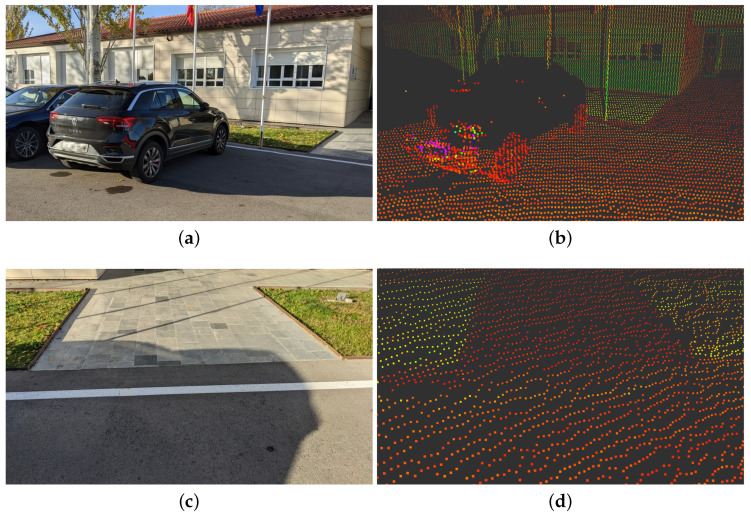
3D LiDAR drawbacks. (**a**) Black car picture. (**b**) Black car in pointcloud. (**c**) Road to pathway picture. (**d**) Road to pathway in pointcloud.

**Figure 4 sensors-23-07633-f004:**
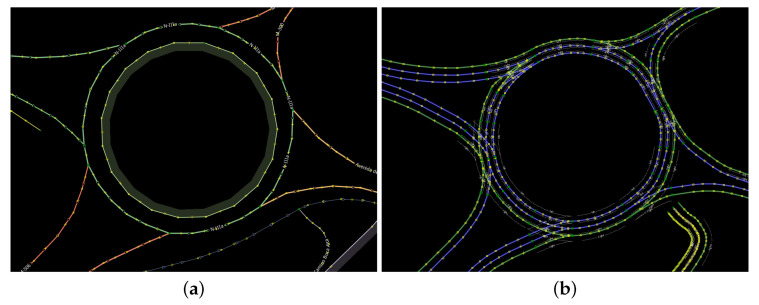
Open source maps. (**a**) Raw OSM. (**b**) Lanelets from OSM.

**Figure 5 sensors-23-07633-f005:**
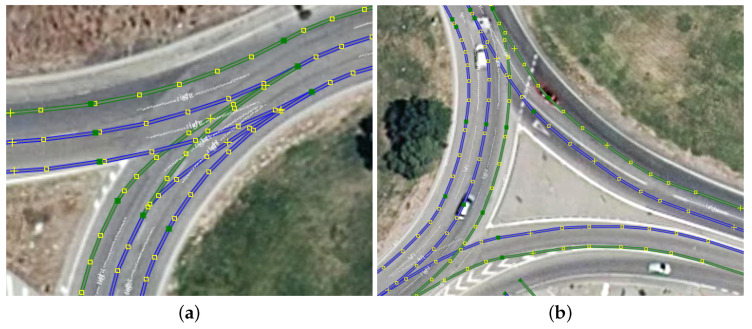
Lanelets from OSM drawbacks. (**a**) Misalignment. (**b**) Wrong number of lanes.

**Figure 6 sensors-23-07633-f006:**
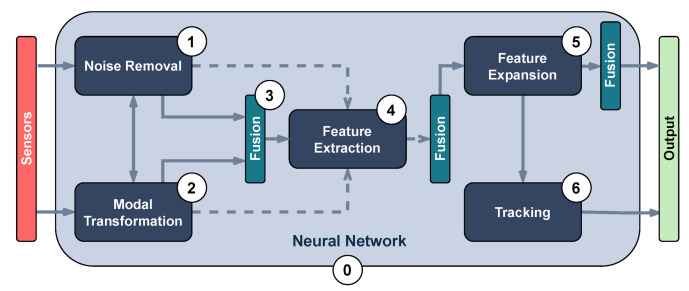
Drivable area estimation architecture breakdown.

**Figure 7 sensors-23-07633-f007:**
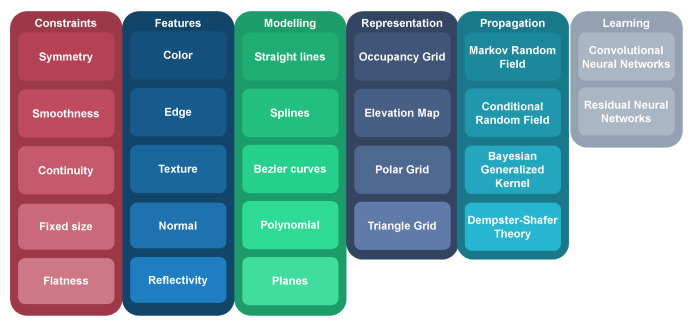
Drivable area estimation taxonomy.

**Table 1 sensors-23-07633-t001:** Dataset comparison.

Dataset	Year	Sensor Type	Sensors	Frames	Map	BM
Waymo Perception [13]	2023	3D LiDAR RGB	5 × Proprietary LiDAR 5 × 1920 × 1280 RGB	390,000	✓	✓
Argoverse 2 [14]	2023	3D LiDAR RGB	2 × VLP-32C 7 × 2048 × 1550 RGB	6,000,000	✓	✓
KITTI ROAD [15]	2015	3D LiDAR RGB	Velodyne HDL-64E 1242 × 375 RGB	579	✗	✓
nuScenes [16]	2020	3D LiDAR RGB RADAR	Velodyne HDL32E 1600 × 900 RGB ARS 408-21	40,000	✓	✓
BDD100K [17]	2020	RGB	1280 × 720 RGB	100,000	✗	✓
CityScapes [18]	2016	Stereo RGB	2048 × 1024 RGB	24,998	✗	✓
KITTI-360 [19]	2021	3D LiDAR Stereo RGB 360º RGB	Velodyne HDL-64E SICK LMS 200 4 × 1408 × 376 RGB	83,000	✗	✓
Semantic KITTI [20]	2019	3D LiDAR	Velodyne HDL-64E	43,552	✗	✓
DIODE [21]	2019	RGB-D	FARO Focus S350	27,858	✗	✗
3DHD CityScenes [22]	2022	3D LiDAR	Trimble Mx8	-	✓	✗
OpenLane V2 [23]	2023	Built on: nuScenes and Argoverse			✓	✓
OMC Benchmark [24]	2021	Built on: nuScenes			✓	✗

**Table 2 sensors-23-07633-t002:** Drivable area segmentation algorithms.

Name	Year	Input	Architecture	Methodology	Output	Contributions	Limitations
BGK [25]	2021	3D LiDAR	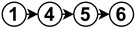	Min–maxing, Bilateral Filtering,BGK, Normal EstimationMulti-frame tracking	Segmented Grid	Generates information fromunobserved areas without LiDAR hits	Computational and algorithmicperformance tied to grid resolution
SNE-RoadSeg [26]	2020	RGB-D	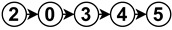	SNE, CNN	Surface NormalsSegmented Image	SNE module that can be plugged intoother CNNs with proven improvement	Surface normal information mightmissclassify sidewalks as road
PLARD [27]	2019	3D LiDARRGB	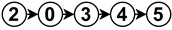	ADT, CNN, FSA	Altitude Difference ImageSegmented Image	Leverages LiDAR to makeRGB data robust to shadows	Requires high-end GPU to perform
USNet [28]	2022	RGB-D	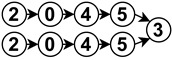	SNE, MEC, Uncertainty Map	Uncertainty ImageProbabilistic SegmentedImage	Good tradeoff between computationaland algorithmic performanceUncertainty map could beused by other modules	Requires high-end GPU to perform
Unsupervised RD [29]	2017	3D LiDARRGB	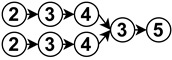	Superpixel, Delaunaytriangulation, MRF,Belief propagation	Superpixel ProbabilisticSegmented ImageProbabilistic SegmentedPointcloud	Robust to illumination changes	Detection of 3D features dictateswhole algorithm performance
Map-Supervised RD [9]	2016	MapRGB	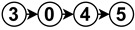	OSM, CNN	Segmented Image	Able to work with orwithout map on inference	Reliant on extrinsic cameracalibration and GPS quality
RBANet [30]	2019	RGB	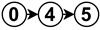	SegNet, BA, RA, CNN	Probabilistic Segmented Image	Residual stages makes algorithminspectable at different stages	Requires high-end GPU to perform
HID-LS [31]	2019	3D LiDARRGB	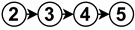	Superpixel, Depth Map,Height Map, Histograms	SuperpixelSegmented Image	Transforms spatially discrete LiDARpointclouds into a continuousand organized structure	Reliant on LiDAR resolutionand parametrization
Curb Detection [32]	2018	3D LiDAR	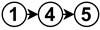	RANSACSliding beam segmentation	Curb Points	Good performance on CPU	Relies on curbs to detect drivable area
LiDAR-Histogram [33]	2017	3D LiDAR	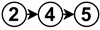	RANSAC, Histogram	LiDAR ImagerySegmented Pointcloud	Detects positive/negative obstacles andestimates road drivability degree	Makes assumptions on voids inLiDAR data which couldmissclassify poorly reflective obstacles
CyberMELD [10]	2020	Map3D LiDARRGB	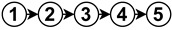	Delaunay triangulation, IPM	Segmented Image	Leverages OSM to deal withmissing features caused byshadows or occlusions	Only validated on single lane two-wayroads Sensitive to OSM errors
RoadNet3 [34]	2019	RGB	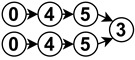	CNN, LSTM	Segmented Image	Reduces feature map resolution with aCNN to achieve high performance	Assumes continuity of drivable area
Double Projection [35]	2021	3D LiDAR	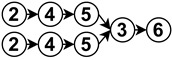	Range Map, Elevation MapForward flood fill	Ground ModelSegmented Pointcloud	Able to deal with rough terrainand offroad situations	Only considers positive obstacles
Pseudo-LiDAR [36]	2022	RGB	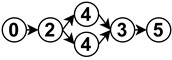	RNN	Pseudo-LiDARSegmented Image	Takes advantage of 3D features butonly requires an RGB camera	Requires additional depth estimatingnetwork to produce pseudo-LiDAR
CLCFNet [37]	2021	3D LiDARRGB	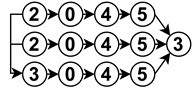	CNN	LiDAR ImagerySegmented PointcloudSegmented Image	Runs on LiDAR only in case RGBis negated	LiDAR imagery over rawpointclouds makes algorithmsensitive to occlusions
Multi-Cue [38]	2016	RGB-D	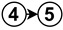	SVR	Curb PointsRoad Boundary ModelSegmented Image	Highest ranking stereo vision algorithmin the KITTI dataset	Uses surface normals to find road boundarieswhich can missclassify sidewalks as road
TRAVEL [39]	2022	3D LiDAR	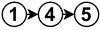	Breadth-first graph search,Tri-grid field	Segmented Pointcloud	Manages to perform on sloped surfacesand rough terrainOne of few non-NN with open-source code	Only focus on traversabledoes not differentiate pathways from road
Road Markings [40]	2022	3D LiDAR	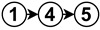	Otsu thresholding,Reflectivity, Line fitting	Lane Boundary Model	Takes advantage of a rarely used data type	Only tested in traffic-free environmentssensitive to parameter tuning
Line Fitting [41]	2019	3D LiDAR	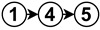	Channel-based segmentationB-spline curve fitting	Road Boundary Model Segmented Pointcloud	Can deal with occlusions Extracted lines can be fused with OSM data	Relies on curb to detect drivable area
YOLOP [42]	2021	RGB	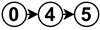	CSPDarknet, SPP, FPN, CNN	Multi-label Segmented Image	Detects opposite lane as non-drivable area Produces drivable area plus lane estimations	Can missclassify gaps in the drivable area as lane-lines
HybridNets [43]	2022	RGB	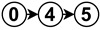	EfficientNet-B3, FPN, CNN	Multi-label Segmented Image	Same as YOLOP while improving computational and algorithmic performance	Limited by camera FOV need to be very close to area to detect it as drivable
Rangenet++ [44]	2019	3D LiDAR	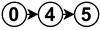	kNN, CNN	Multi-label Segmented Pointcloud	Offers open-source trained pointclouds tackles over-segmentation in postprocessing	Requires high-end GPU to perform Hard-linked to LiDAR specifics in training
SpatioTemporal CRF [45]	2017	3D LiDAR	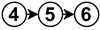	CRF	Segmented Pointcloud	Robust to changes in slope and temporarily obstructed areas	No quantitative results Implementation details bare in the publication
Urban Road Filter [46]	2021	3D LiDAR	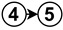	Channel-based segmentation	Curb Points Road Boundary Model	Works in real-time on CPU	To detect road it requires the curb to be visible and the sidewalk to be smooth
Evidential Grids [47]	2015	Map 3D LiDAR	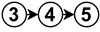	DST	Evidential Grid	Deals very well with noisy input data	Requires previously segmented data Assumes that the map used is high-fidelity
RoadSLAM [48]	2019	Map 3D LiDAR	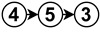	OSM, B-Spline, UKF, Channel-based segmentation, GraphSLAM	Road Boundary Model	Is able to produce road estimations in areas out of sensor reach	Does not work on complex road geometries such as junctions
SemanticDepth [49]	2019	RGB	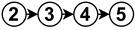	CNN	Pseudo-LiDAR Segmented Image	Robust to occlusions Specifically designed to work without lane-lines	Computationally expensive

**Table 3 sensors-23-07633-t003:** Algorithms taxonomy.

Name	Constraints					Features					Model				Representation				Propagation				Learning	
	**Sym**	**Smo**	**Con**	**FS**	**Flat**	**C**	**E**	**T**	**N**	**R**	**SL**	**Sp**	**P**	**Pl**	**OG**	**EM**	**PG**	**TG**	**MRF**	**CRF**	**BGK**	**DST**	**CNN**	**RNN**
BGK [25]	–	–	–	–	–	–	✓	–	✓	–	–	–	–	–	–	✓	–	–	–	–	✓	–	–	–
SNE-RoadSeg [26]	–	–	✓	–	–	✓	–	–	✓	–	–	–	–	–	–	–	–	–	–	–	–	–	✓	–
PLARD [27]	–	–	✓	–	–	✓	–	✓	–	–	–	–	–	–	–	–	–	–	–	–	–	–	✓	–
USNet [28]	–	–	✓	–	–	✓	–	✓	–	–	–	–	–	–	–	–	–	–	–	–	–	–	✓	–
Unsupervised RD [29]	–	–	–	–	✓	✓	✓	✓	✓	–	–	–	–	–	–	–	–	–	✓	–	–	–	–	–
Map-Supervised RD [9]	–	–	✓	–	–	✓	–	✓	–	–	–	–	–	–	–	–	–	–	–	–	–	–	✓	–
RBANet [30]	–	–	✓	–	–	✓	✓	✓	–	–	–	–	–	–	–	–	–	–	–	–	–	–	–	✓
HID-LS [31]	–	–	–	–	✓	✓	✓	–	✓	–	–	–	–	–	–	✓	–	–	–	–	–	–	–	–
Curb Detection [32]	✓	✓	–	✓	–	–	✓	–	–	–	–	–	✓	–	–	–	–	–	–	–	–	–	–	–
LiDAR-Histogram [33]	–	–	–	–	✓	–	✓	–	–	–	✓	–	–	–	–	–	–	–	–	–	–	–	–	–
CyberMELD [10]	–	–	✓	–	✓	–	✓	–	–	–	✓	–	–	–	–	–	–	–	–	–	–	–	–	–
RoadNet3 [34]	–	–	✓	–	–	✓	–	–	–	–	–	–	–	–	–	–	–	–	–	–	–	–	✓	–
Double Projection [35]	–	✓	✓	–	–	–	✓	–	–	–	–	–	–	–	✓	✓	–	–	–	–	–	–	–	–
Pseudo-LiDAR [36]	–	–	✓	–	–	✓	✓	✓	–	–	–	–	–	–	–	–	–	–	–	–	–	–	–	✓
CLCFNet [37]	–	–	✓	–	–	✓	–	✓	–	–	–	–	–	–	–	–	–	–	–	–	–	–	✓	–
Multi-Cue [38]	–	–	✓	–	–	✓	✓	–	✓	–	–	–	–	✓	–	–	–	–	–	–	–	–	–	–
TRAVEL [39]	–	–	✓	–	✓	–	✓	–	✓	–	–	–	–	–	–	–	–	✓	–	–	–	–	–	–
Road Markings [40]	✓	–	–	–	–	–	–	–	–	✓	✓	–	–	–	–	–	–	–	–	–	–	–	–	–
Line Fitting [41]	–	–	✓	–	–	–	✓	–	–	–	–	✓	–	–	–	–	✓	–	–	–	–	–	–	–
YOLOP [42]	–	–	✓	–	–	✓	–	✓	–	–	–	–	–	–	–	–	–	–	–	–	–	–	✓	–
HybridNets [43]	–	–	✓	–	–	✓	–	✓	–	–	–	–	–	–	–	–	–	–	–	–	–	–	✓	–
Rangenet++ [44]	–	✓	✓	–	–	–	–	–	–	–	–	–	–	–	–	–	–	–	–	–	–	–	✓	–
SpatioTemporal CRF [45]	–	–	–	–	–	–	–	–	–	–	–	–	–	–	–	✓	–	–	–	✓	–	–	–	–
Urban Road Filter [46]	–	✓	✓	–	✓	–	✓	–	–	–	–	–	–	–	–	–	–	–	–	–	–	–	–	–
Evidential Grids [47]	–	–	–	–	–	–	–	–	–	–	–	–	–	–	✓	–	–	–	–	–	–	✓	–	–
RoadSLAM [48]	–	✓	✓	–	–	–	–	–	✓	–	–	✓	–	–	–	–	–	–	–	–	–	–	–	–
SemanticDepth [49]	–	–	–	–	–	–	–	–	–	–	–	–	–	–	–	–	–	–	–	–	–	✓	–	–

**Table 4 sensors-23-07633-t004:** Algorithm reported performance.

Name	Time [ms]	Pre.	Rec.	F1	GPU	Dataset	Code
BGK [25]	45	98.66	72.72	83.73	✗	SemKITTI	✗
SNE-RoadSeg [26]	80	96.90	96.61	96.75	✓	KITTI	✓
PLARD [27]	160	96.79	96.86	96.83	✓	KITTI	✓
USNet [28]	22	96.51	97.27	96.89	✓	KITTI	✓
Unsupervised RD [29]	-	83.97	91.83	87.72	✗	KITTI	✗
Map-Supervised RD [9]	280	86.01	89.66	87.80	✓	KITTI	✗
RBANet [30]	160	95.14	97.50	96.30	✓	KITTI	✗
HID-LS [31]	250	92.52	93.71	93.11	✗	KITTI	✗
Curb Detection [32]	12	87.64	89.28	86.98	✗	Proprietary	✗
LiDAR-Histogram [33]	100	93.06	88.41	90.67	✗	KITTI	✗
CyberMELD [10]	50	95.94	91.30	93.56	✗	KITTI	✓
RoadNet3 [34]	16	88.12	90.06	89.08	✓	KITTI	✗
Double Projection [35]	77	95.91	99.28	95.00	✗	SemKITTI	✗
Pseudo-LiDAR [36]	460	97.30	97.54	97.42	✓	KITTI	✗
CLCFNet [37]	23	96.38	96.39	96.38	✓	KITTI	✗
Multi-Cue [38]	2500	84.95	88.55	86.71	✗	KITTI	✗
TRAVEL [39]	19	90.00	96.70	93.10	✗	SemKITTI	✓
Road Markings [40]	-	97.04	94.03	95.51	✗	Proprietary	✗
Line Fitting [41]	36	94.95	94.95	94.95	✗	Proprietary	✗
RoadSLAM [48]	-	87.00	92.00	89.43	✗	Proprietary	✗
YOLOP [42]	43	mIoU: 91.5			✓	BDD100K	✓
HybridNets [43]	37	mIoU: 90.5			✓	BDD100K	✓
Rangenet++ [44]	76	mIoU: 91.8			✓	SemKITTI	✓
SpatioTemporal CRF [45]	147	only qualitative			✓	KITTI	✗
Urban Road Filter [46]	15	only qualitative			✗	Proprietary	✓
Evidential Grids [47]	-	only qualitative			✗	Proprietary	✗
SemanticDepth [49]	637	MAE: 0.48 m			✗	Proprietary	✗

## Data Availability

Not applicable.

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
