# Peer review of "Recent Developments on Drivable Area Estimation: A Survey and a Functional Analysis"

_sensors, 2023, doi:10.3390/s23177633_

Round 1

Reviewer 1 Report

The survey presented a set of segmentation algorithms regards to drivable area estimation and the study of the existing relevant datasets to assess the performance of these algorithms.

Minor comments

In Table 2. Architecture numbers need claryfication or a link to chapter 4.1. fig 6.

Reviewer 2 Report

Drivable area estimation has been extensively studied over recent years, and can be classified in many different categories based on technologies, environment conditions, traffic conditions, online/offline, 2D/3D, with/without map, working range,... This review manuscript only gives a basic overview in this area and would include more studies in the literature. 

This manuscript is a review work. It introduces a summarized review of studies in determining the drivable area for autonomous driving applications. As this is a review work, it is not an original research work. Eventhough, the topic is significant in its research field. However, the main concern with the manuscript is not the topic, but the quality of the content.

As a review manuscript, the readers would expect an insight classification, analysis and comparison of the works published in the field.

For the classification, the authors have categorized the mentioned papers in drivable area perception or estimation, sensor types of RGB camera or LIDAR, and algorithms. As drivable area estimation is a large topic that includes many smaler topics such as road detection, lane detection, traffic sign detection and recognition, obstacle detection, traffic prediction. The author should also give a definition for what can be considered as "drivable area" first. For example, is it an area that is accessible physically for a vehicle, or an area that is not prohibited by traffic signs, or an area that is not risked by accidents?

For the analysis, following the mentioned papers, the authors did not give insight analysis for pros and cons, the limitations and functioning conditions for each of them. From the current manuscript, it is not possible to answer what technologies, approaches are attracting or worth for further research and applications.

For the comparison, the authors have given only a basic comparison of the mentioned works.

The authors should consider to give a clearer topic definition and problem boundary, to mention more works in the literature, include more features in the comparisons and have more insight analysis of the mentioned works.

The conclusions are supported by the data. However, the conclusions are basic and are NOT very useful for someone willing to debut a study in this field.

Reviewer 3 Report

This paper provides the state of the art on drivable area estimation. It introduces a novel architecture breakdown that fits learning-based and non-learning-based techniques and allows the analysis of a set of impactful and recent drivable area algorithms. Complimentary information for practitioners is also provided: (i) an assessment of the influence of modern sensing technologies in the task under study; (ii) a selection of relevant datasets for evaluation and benchmarking purposes. In general, the paper is well written and organized. However, there are still some detail questions on the system descriptions and the writing.

1.     In section 2.3, it is said that “This source of information can be useful to tackle the drivable area estimation task since it contains geospatial limits to the road”, it is suggested that some literature which use this method should be provide to support this viewpoint.

2.     In section 2.3, it is said that errors introduced in OSM make it cannot be a complete and standalone solution, is “errors introduced in OSM” refers to the error in lane markings in the following example? It is suggested to provide more detailed and accurate descriptions.

3.     In section 4.1, feature extraction may occur before or after fusion, and this relationship cannot be shown in Fig.6, it is suggested to be modified.

4.     In section 4.1, the fusion module refers to multi-sensor fusion. The inputs of fusion module shown in Fig.6 lead to ambiguity easily, it is suggested to be modified.

5.     In section 4.1, texture of a road needs to be extracted from the original sensor data through feature extraction, should it belong to a high-level feature?

6.     The article should give a complete architecture of the collision avoidance model.

7.     In table 2, the architecture of each algorithm is unclear. For example, Spatio Temporal CRF just has three module, but we can not get what do these three module mean. It is suggested to be modified, for example, different colors or shapes for different module.

8. It is recommended to provide some conclusions to describe the testing result of these algorithms after Table 3. 

The writing is generally acceptable.

Round 2

Reviewer 2 Report

My suggestions in the last round have been addressed in the updated version. The manuscript has been largely improved. I have no further comment.

Reviewer 3 Report

The revised manuscript can be accepted.